# Emerging Role of Immune Checkpoint Inhibitors in Hepatocellular Carcinoma

**DOI:** 10.3390/medicina55100698

**Published:** 2019-10-17

**Authors:** Vito Longo, Oronzo Brunetti, Antonio Gnoni, Antonella Licchetta, Sabina Delcuratolo, Riccardo Memeo, Antonio Giovanni Solimando, Antonella Argentiero

**Affiliations:** 1Medical Thoracic Oncology Unit, IRCCS Istituto Tumori “Giovanni Paolo II” of Bari, 70124 Bari, Italy; vito.longo79@tiscali.it; 2Medical Oncology Unit, IRCCS Istituto Tumori “Giovanni Paolo II” of Bari, 70124 Bari, Italy; dr.oronzo.brunetti@tiscali.it (O.B.); antoniogiovannisolimando@gmail.com (A.G.S.); 3Medical Oncology Unit, “S. Cuore di Gesù” Hospital, 73014 Gallipoli, Italy; drgnoni.antonio@libero.it (A.G.); antonellalicchetta@libero.it (A.L.); 4Scientific Direction, IRCCS Istituto Tumori “Giovanni Paolo II” of Bari, 70124 Bari, Italy; delcuratolo.sa@gmail.com; 5Department of Emergency and Organ Transplantation, University of Bari “Aldo Moro”, 70124 Bari, Italy; drmemeo@yahoo.it; 6Department of Biomedical Sciences and Human Oncology, Section of Internal Medicine “G. Baccelli”, University of Bari “Aldo Moro”, Bari 70124, Italy

**Keywords:** hepatocellular carcinoma, immune checkpoint inhibitors, HCC, pembrolizumab, nivolumab, immune microenvironment, targeted therapies

## Abstract

Hepatocellular carcinoma is the most common primary liver cancer and the fourth leading cause of cancer death worldwide. A total of 70–80% of patients are diagnosed at an advanced stage with a dismal prognosis. Sorafenib had been the standard care for almost a decade until 2018 when the Food and Drug Administration approved an alternative first-line agent namely lenvatinib. Cabozantinib, regorafenib, and ramucirumab also displayed promising results in second line settings. FOLFOX4, however, results in an alternative first-line treatment for the Chinese clinical oncology guidelines. Moreover, nivolumab and pembrolizumab, two therapeutics against the Programmed death (PD)-ligand 1 (PD-L1)/PD1 axis have been recently approved for subsequent-line therapy. However, similar to other solid tumors, the response rate of single agent targeting PD-L1/PD1 axis is low. Therefore, a lot of combinatory approaches are under investigation, including the combination of different immune checkpoint inhibitors (ICIs), the addition of ICIs after resection or during loco-regional therapy, ICIs in addition to kinase inhibitors, anti-angiogenic therapeutics, and others. This review focuses on the use of ICIs for the hepatocellular carcinoma with a careful assessment of new ICIs-based combinatory approaches.

## 1. Introduction

Hepatocellular carcinoma (HCC) is the sixth most common cancer and the fourth leading cause of cancer death worldwide as stated in reports as of 2018. Chronic hepatitis C virus (HCV) or chronic hepatitis B virus (HBV) infections, alcohol abuse, and non-alcoholic steatohepatitis are the main risk factors [1,2]. An association between type 2 diabetes mellitus and HCC has also been reported [3]. Treatment for the early stage includes hepatectomy, liver transplant, hepatic transarterial chemoembolization (TACE), and radiofrequency ablation (RFA). 

Nevertheless, 70–80% of patients cannot benefit from such opportunities because they are diagnosed at an advanced stage and can receive only palliative care. Sorafenib has been the standard choice for a decade in advanced HCC, even if other Tyrosine Kinase Inhibitors (TKIs) as well [4] are characterized by possible primary resistance or acquired resistance [5]. Recently, lenvatinib showed similar results in terms of survival in a non-inferiority randomized trial study considering the same subset of patients [6]. Meanwhile, cabozantinib, regorafenib, and an anti-Vascular Endothelial Growth Factor Receptor (VEGFR) 2 molecules namely ramucirumab exhibited promising results in the second-line setting [7,8,9]. However, the response rate of TKIs in HCC remains low and new treatment approaches are needed. As known, cancers cells are able to evade immunosurveillance, promoting tumor growth and progression through the activation of different immune checkpoint pathways. Monoclonal antibodies targeting immune checkpoints have determined an enormous breakthrough in cancer therapeutics, promoting the immune-mediated elimination of tumor cells. Among these, Programmed death (PD)-ligand 1 (PD-L1)/PD1 and Cytotoxic T-Lymphocyte Antigen (CTLA)-4 inhibitors are actually used in several malignancies, whereas molecules able to disrupt other co-inhibitory signalling pathways are under investigation, such as T cell immunoglobulin and immune-receptor tyrosine-based inhibitory motif domain (TIGIT), Lymphocyte activation gene-3 (LAG3), and T cell immunoglobulin containing the mucin domain 3 (TIM-3). In the era of immunotherapy, immune checkpoint inhibitors (ICIs) have also been tested for HCC patients [10], in particular, nivolumab and pembrolizumab result in approved for second-line therapy. However, similar to other gastrointestinal malignancies, [11] the HCC response rate of ICIs as a monotherapy is low, therefore new combinatory approaches comprising TKIs, the addition of different ICIs, anti-angiogenic therapeutics, locoregional therapy, other kinase inhibitors, chemotherapy, and other drugs are currently under intensive investigations.

## 2. From Liver Immune System to HCC Immune Disorders

The liver is an organ with a specific blood supply that influences its immune microenvironment. In particular, 75% of the blood vascularises the liver through the portal vein that drains into smaller diameter structures called sinusoids. Therefore, a great number of antigens are in contact through the liver sinusoidal endothelial cells (LSECs) with the liver immune microenvironment, consisting in hepatic stellate cells (h-SCs) [12], Kupffer cells [13], fibroblasts [14], dendritic cells (DCs) [15], and lymphocytes [16]. The tumour microenvironment actively participates in drug-resistance acquisition in solid tumours [17,18,19]. Both lymphocytes and DCs present multiple subtypes with different pro-tumorigenic functions [20,21] including both the response to pathogenic non-self antigens and the tolerance to self-antigens. As far as immune-tolerance is concerned, the hepatic microenvironment shows a high expression of the hepatocyte growth factor and colony-stimulating factor 1, which promote a tolerigenic phenotype that is required to overcome autoimmune mechanisms due to antigenic hyperstimulation coming from the bowel [22]. In addition, HCV and HBV infections lead to frequent chronic inflammatory liver insult resulting in a deregulation of T cell activities with an increase of the expression of ICIs [23]. At the same time, HCC patients exhibit a more immunosuppressed microenvironment of the liver [24] with an increased number of regulatory T cells (Tregs), tumor-associated macrophages (TAMs), and myeloid-derived suppressor cells (MDSC) which correlate with tumor progression and poor prognosis [25,26]. Given the tight correlation between the endothelial and epithelial function in immune response modulation against different cancer types [27,28,29], extensive investigation in HCC patients pinpoint LSECs as immune tolerance inducer of CD8-positive T cells to tumor-associated antigens, inducing T-regs, and also increase PD-L1 expression which correlate with recurrence after surgery and poor prognosis in advanced HCC (Figure 1) [30].

## 3. Checkpoint Inhibitors

Efficacy and safety of ICIs have been evaluated in several trials in HCC (Table 1). In September 2017, based on the results of a phase I/II nonrandomized multi-institution study (CheckMate 040), the Food and Drug Administration (FDA) approved a full human immunoglobulin G4 monoclonal antibody direct to PD1, namely Nivolumab for HCC patients who progressed on or after sorafenib. The trial included 48 patients in a dose-escalation phase and 214 patients in a dose-expansion phase. Among the dose-escalation phase, nivolumab resulted in a disease control rate of 55% and an objective response rate of 10%, with a median OS (OS) of the entire cohort of 7.6 months that increase to 9.8 months in sorafenib-naïve patients [31]. CheckMate 459, a multi-centre, randomized, phase III trial assessed nivolumab compared with sorafenib as a first-line treatment in 1009 HCC patients, results are currently in process (NCT02576509) [32], nonetheless, a press release reported that the study did not achieve statistical significance for its primary endpoint of OS. More recently, the FDA granted accelerated approval for pembrolizumab, another antibody targeting PD-1, for patients with HCC previously treated with sorafenib. In the nonrandomized, open-label, phase II KEYNOTE-224 trial including 104 patients who progressed on or were intolerant to sorafenib, pembrolizumab demonstrated an objective response of 17% with one complete response and 17 partial response (PR), while 46 patients experienced a stable disease (SD). The median progression-free survival (PFS) was 4.9 months and the median OS was 12.9 months [33]. However, similar to the phase III CheckMate 459, the phase 3 Keynote-240 trial comparing pembrolizumab to a placebo in second-line HCC did not meet its coprimary endpoints of OS and PFS [34]. The monoclonal antibody anti-Cytotoxic T-Lymphocyte Antigen (CTLA) 4, tremelimumab was assessed in a phase II multi-center clinical trial including 20 patients with advanced HCC from hepatitis C viral aetiology. The infusion of tremelimumab at the dose of 15 mg/Kg IV every 90 days resulted in 18% of PR and a 60% of SDwith a PFS of 6.48 months (95% CI 3.95–9.14). Interestingly, a significant drop in the viral load was observed [35]. In addition, at the recent ESMO 2019, data concerning the use of ICIs in first-line treatment was examined. The CheckMate 459, a Phase 3 Study comparing Nivolumab (NIVO) vs. Sorafenib has shown amedian OS of 16.4 months for nivolumab and 14.7 months for SORAFENIB, with higher ORR and lower toxicities [36].

Abbreviation: CP: Child-Pugh; CTLA4: Cytotoxic T-Lymphocyte Antigen-4; DCR: disease control rate; HCC: hepatocellular carcinoma; mOS: median Overall survival; mPFS: median Progression Free Survival; ORR: overall response rate; PD-1: Programmed death 1; PD-L1: Programmed death-ligand 1; PR: partial response; SD: stable disease; TACE: transarterial chemoembolization; TKI: tyrosine Kinase Inhibitor.

Unfortunately, the response rate of single agent ICI remains low, differently from the circulating CD8+ T-cells that increased after ICIs treatment, none activity enhancement have been observed for intrahephaticCD8+ T-cells. The combination of anti-CTLA4 and antibody targeting the PD1/PD-L1 axis are also under investigation, based on preclinical studies demonstrating that the 2 pathways are not overlapping, indeed it seems that the combination has a synergistic effect able to reverse the refractoriness of intrahepatic CD8+ T-cells [37]. The combination of the anti-CLA4 antibody ipilimumab and nivolumab is currently assessed in patients undergoing hepatic resection as a neoadjuvant treatment (NCT03682276, NCT03510871) [38,39]. Recently, monthly tremelimumab 75 mg in combination with the anti-PD-L1 durvalumab 1500 mg for 4 doses followed by monthly durvalumab 1500 mg monotherapy until progression has been assessed in patients with advanced HCC who had received at least one prior therapy. Disease control rate is 60% with a median PFS of 7.8 months (95% CI 2.6 to 10.6 months) and median OS of 15.9 (95% CI 7.1 to 16.3 months).

Combined immune checkpoint inhibition (ICI) with tremelimumab and durvalumab in patients with advanced hepatocellular carcinoma (HCC) or biliary tract carcinomas (BTC) [40].

Instead, in the adjuvant setting, for patients who have undergone a remedial resection, toripalimab a anti-PD-1 antibody has being assessed in the JUPTER-04 trial with the primary end-point consisting in recurrence-free survival [41].

## 4. Combined Approaches with Checkpoint Inhibitors

Despite the fact that the impact of ICIs on the treatment of malignancies is unprecedented, unlike melanoma and non-small cell lung cancer, the response rate in HCC remains low. In regards to this, as well as for other malignancies, researchers are assessing combined approaches to increase the efficacy of ICIs [42]. The combination of other therapeutics with ICIS is able to modify the immune microenvironment of the tumor, up-regulating T cells with effector functions, and decreasing the immunosuppressive cells expression, and so changing a cold tumor into a hot one The combination of ICIs with anti-Vascular Endothelial Growth Factor (VEGF) therapy is a major approach under investigation for HCC patients using the immunomodulatory effects of an anti-VEGF drug as a means of decreasing CD4+ regulatory T-Lymphocytes and MDSCs as well as the activation and differentiation of DCs [43,44]. The combination of atezolizumab, a fully humanized, engineered monoclonal antibody of IgG1 isotype anti-PD-L1 and bevacizumab, a monoclonal antibody anti-VEGF, resulted in 61% partial response among 21 HCC patients with a relatively positive tolerability characterized by 35% of the subjects experiencing grade III/IV adverse events [45]. The combination of atezolizumab plus bevacizumab is also being assessed in a phase III trial, open-label, multi-center, randomized study, with sorafenib in the control arm, in pts with locally advanced or metastatic and/or unresectable HCC (NCT03434379) [46]. Another randomized phase III EMRALD-2 (NCT03847428) [47] is assessing the role of another ICIs targeting PD-L1, namely durvalumab, in addition to bevacizumab in the adjuvant setting.

Ater sorafenib, other TKIs have been approved for HCC patients [6,9]. Interestingly, evidence suggests that these small molecule inhibitors could improve tumor immunogenicity through the increase of antigen expression and the activation of cytotoxic activity of CD8+T-cells. Several trials are exploring the combination of ICIs with TKIs approved for HCC patients, such as regorafenib, cabozantinib, and levatinib. Preliminary results of a phase 1b trial of lenvatinib with pembrolizumab demonstrated that among the 13 patients treated, 6 (46%) achieved a PR and 6 (46%) a SD [48]. A phase 3 assessing this combination for the first-line treatment of patients with advanced HCC is ongoing [49]. Two cohorts were added to the phase I-IINCT01658878clinical trial with the aim to assess the tolerability and the effectiveness of nivolumab in combination with sorafenib and with cabozantinib [50].

Another combined approach under investigation is the addition of ICIsto locoregional therapies, in particular, the post-transarterial chemoembolization (TACE), which is associated with antigen release and the exposure of damage-associated molecular patterns. A phase I trial concerning HCC patients treated with TACE, radiofrequency ablation, or cryoablation in combination with tremelimumb, showed 23.5% (4 of 17 patients) of PR [51]. A tumor biopsy at 6 weeks showed an increase of CD8+ T-cells infiltration in patients who showed a clinical benefit. Others trials combing ICIs with locoregional therapy are ongoing: pembrolizumab with TACE (phase I/II, NCT03397654) [52], nivolumab with TACE (phase I, NCT03143270) [53], pembrolizumab with yttrium-90 radioembolization (phase II, NCT03099564) [54], and nivolumab with yttrium-90 radioembolization (phase II, NCT03812562) [55], among others.

Even if HCC is known as a malignancy which is highly refractory to chemotherapy, based on the results of the EACH study, FOLFOX4 has been recently recommended as a clinical practice guideline by the China FDA 8.6% of partial response and 38.6% of stable disease have been reported with a median OS of 5.7 months [56]. Interestingly, similar to other chemotherapeutic agents, oxaliplatin can induce an antitumor immune response, activating DCs, promoting the antitumor CD4+T cells phenotype, and by the down-regulation of MDSC and regulatory T cells. Regarding this factor, a monoclonal antibody directed againstPD-1 namely SHR-1210 combined with FOLFOX4 is under investigation in Chinese patients with advanced HCC (NCT03092895) [57].

Finally, in addition to the co-inhibitory receptors CTLA-4 and PD-1 other co-inhibitory molecules have been described, including TIGIT, LAG3, and TIM-3 [27]. The latter has been shown to be involved in HCC progression, with high infiltration of TIM-3 positive cells correlating with poor prognosis. The combination of anti-PD-L1/anti-PD1 therapy with therapeutics targeting TIM-3 (NCT03099109) [58] and LAG-3 (NCT03005782and NCT01968109) [59,60] is under investigation.

## 5. Conclusions and Future Directions

As discussed in this manuscript, ICIs are under intensive investigation for HCC patients, as well as other malignancies, ICIs arealready approved for lines subsequent to the first, such as nivolumab and pembrolizumab. However, in the same way as in other solid tumors, the response rate is low, therefore several strategies to improve the efficacy of ICIs are under investigation. Several prospective clinical trials are assessing the efficacy and safety of PD1 blockade in combination with CTLA-4 blockade, since this combination should enhance the results of single agent anti-PD1. This combination should enhance OS and long term survival rates of both localized and metastatic HCC. In any case, it will take several years before obtaining not only results from these data, but also discovering predictive factors able to select patient candidates to ICIs therapy. The combinations of different ICIs with anti-angiogenic therapeuticsor of kinase inhibitors, are now being tested in HCC as well, with the aim to improve the anticancer effectiveness and prolong cancer survival. Interestingly, a unique and promising combinatory approach that is being assessed in HCC patients is the addition of ICIs to locoregional therapy.

In the next years, several preclinical and clinical studies on HCC biology and treatment will address tailored therapies and combinations for these malignancies. In this scenario, immune oncology will have a special role in HCC. Even if, currently, the immunotherapy remains a non-curative treatment for HCC patients, the better tolerability of ICIs may display a crucial role in the improvement of the quality of life.

## Figures and Tables

**Figure 1 medicina-55-00698-f001:**
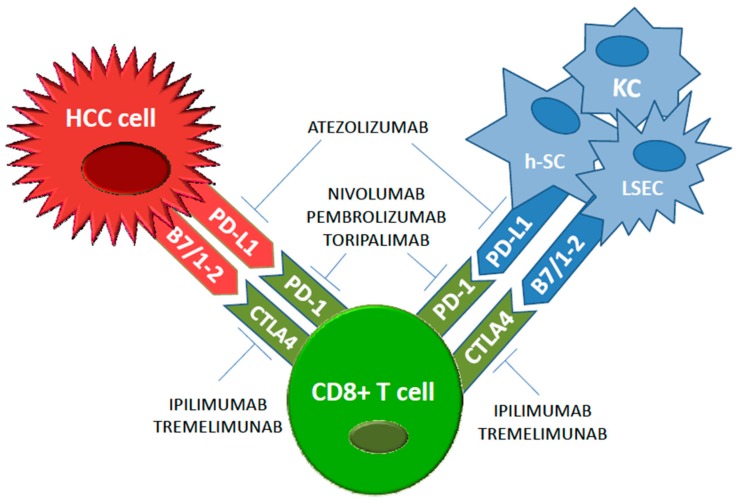
Main immune checkpoints and relative immune checkpoint inhibitors in hepatocellular carcinoma (HCC).

**Table 1 medicina-55-00698-t001:** Clinical trials of immune checkpoint inhibitors in HCC.

Treatments	Mediators	Design	Setting	Primary Outcomes (Finished/Ongoing)	Reference
NIVOLUMAB	Anti-PD1	Dose-escalation Dose-expansion phase trial	Second line in sorafenib pretreated patients	DCR: 55%; ORR: 10%, mOS 9.8 months (finished)	31
NIVOLUMAB vs. SORAFENIB	Anti-PD1	Phase III trial	First line treatment	mOS (ongoing, preliminary negative results)	32
PEMBROLIZUMAB	Anti-PD1	Phase II trial	Second line in sorafenib pretreated patients	ORR: 17%; mPFS: 4.9 months; mOS: 12.9 months (finished)	33
PEMBROLIZUMAB vs. PLACEBO	Anti-PD1	Phase III trial	Second line	Improved OS (HR: 0.78; *p* = 0.0238) and PFS (HR: 0.78; *p* = 0.0209)—does not meet significance per the prespecified statistical plan (ongoing, preliminary negative results)	34
TORIPALIMAB	Anti-PD1	Phase II trial	Adiuvant setting	Recurrence-freesurvival (ongoing)	41
TREMELIMUMAB	Anti-CTLA4	Phase II trial	Pretreated advanced HCC from hepatitis C viral etiology	18% of PR and a 60% of SD (finished)	35
NIVOLUMAB plus IPILIMUMAB	Anti-CLA4 plus Anti-PD1	Phase I-II trial	Neoadjuvant treatment	Delay to surgery Incidence of treatment-emergent adverse events (ongoing)	38
NIVOLUMAB plus IPILIMUMAB	Anti-CLA4 plus Anti-PD1	Phase II trial	Neoadjuvant treatment	The percentage of subjects with tumor shrinkage after drug treatment study (ongoing)	39
ATEZOLIZUMAB plus BEVACIZUMAB	Anti-PD1 plus antiangiogenic drug	Phase II trial	First line treatment	61% PR with a relatively positive tolerability (finished)	45
ATEZOLIZUMAB plus BEVACIZUMAB vs. SORAFENIB	Anti-PD1 plus antiangiogenic drug	Phase III trial	Metastatic and/or unresectable HCC (first line)	OS/PFS (ongoing)	46
DURVALUMAB (D) plus BEVACIZUMAB (B) vs. D vs. placebo	Anti-PDL1 plus antiangiogenic drug	Phase III trial	Adjuvant setting	Recurrence-free survival (ongoing)	47
LENVATINIB plus PEMBROLIZUMAB	Anti-PD1 plus TKI	Phase 1b trial	unresectable HCC (first line)	46% of PR and 46% of SD (finished)	48
LENVATINIB (L) plus PEMBROLIZUMAB vs. L	Anti-PD1 plus TKI	Phase III trial	Advanced HCC (first line)	PFS/OS (ongoing)	49
NIVOLUMAB (N) vs. IPILIMUMAB (I) + N vs. SORAFENIB N vs. CABOZANTINIB (C) + N vs. SORAFENIB (CP−A) vs. N+C+I	Anti-PD1 and Anti-CTLA4 plus TKI	Phase I-II trial	CP-A HCC CP-B HCC	Safety and Tolerability (ongoing)	50
TACE, radiofrequency ablation, or cryoablation plus TREMELIMUMAB	Anti-CTLA4 plus interventional radiological procedures	Phase 1b trial	Locally advanced HCC	23.5% of PR (finished)	56
PEMBROLIZUMAB plus TACE	Anti-PD1 plus interventional radiological procedures	Phase I-II trial	Locally advanced HCC	Safety and Tolerability (ongoing)	52
NIVOLUMAB plus TACE	Anti-PD1 plus interventional radiological procedures	Early phase I trial	Locally advanced HCC	Safety and Tolerability (ongoing)	53
PEMBROLIZUMAB plus yttrium90 radioembolization	Anti-PD1 plus interventional radiological procedures	Early phase I trial	Locally advanced HCC	PFS (ongoing)	54
NIVOLUMAB plus yttrium90 radioembolization	Anti-PD1 plus interventional radiological procedures	Early phase I trial	Locally advanced HCC	Recurrence rate (ongoing)	55
SHR-1210 + ApatinibSHR-1210 + FOLFOX4 or GEMOX regimen	Anti-PD1 plus chemotherapy or TKI	Phase II trial	Advanced Primary Liver Cancer	Safety and Tolerability (ongoing)	57

Abbreviation: CP: Child-Pugh; CTLA4: Cytotoxic T-Lymphocyte Antigen-4; DCR: disease control rate; HCC: hepatocellular carcinoma; mOS: median Overall survival; mPFS: median Progression Free Survival; ORR: overall response rate; PD-1: Programmed death 1; PD-L1: Programmed death-ligand 1; PR: partial response; SD: stable disease; TACE: transarterial chemoembolization; TKI: tyrosine Kinase Inhibitor.

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
