# Peer review of "Emerging Role of Immune Checkpoint Inhibitors in Hepatocellular Carcinoma"

_medicina, 2019, doi:10.3390/medicina55100698_

Round 1
Reviewer 1 Report
In this review, the authors investigated a variety of concomitant medications that could be counteracted because the efficacy of monotherapy with immune checkpoint inhibitors (ICIs) was not significant. This reviewer would like to raise some issues for the revision.
Paragraph 1: There is a very little description of introduction to ICIs. It would be better if the introduction part should include a comprehensive description of ICIs for chemo-therapeutics. Line 82: 7,6 months, 7.6 months. There are needing the same revisions throughout the article. line 91-92: The results of the PFS and OS on the untreated (or with another option for treatment) comparison should be presented. Paragraph 3: No information was given about the survival rate following CTLA-4. Paragraph 4: There is a lack of etiology information for the subjects in efficacy by combination with ICIs. It is advisable to state what pre-treatment the patient has undergone at a certain stage. Lines 145-152: No information on the clinical outcomes following FOLFOX4. It would be better if paragraph 4 will be divided into two groups. The first group is currently undergoing clinical progress and that can provide results for PFS, OS, and so on. The other group is under preclinical or early stage of clinical trials. Please discuss it in the conclusion or optimal part. Even if it's a palliative care targeting terminal cancer patients, it's always debated point whether it's judged that prolonging life about 2 months can have significant clinical significance.
Author Response
October 07th, 2019
Prof. Dr. Edgaras Stankevičius
Editor-in-Chief
Dear Prof. Dr. Edgaras Stankevičius,
Please find attached our revised manuscript entitled: “Emerging role of Immune Checkpoint Inhibitors in Hepatocellular Carcinoma”.
To the best of our knowledge this is an interesting review reporting the role of immune checkpoint inhibitors in hepatocellular carcinoma.
We prepared the manuscript with the revision required and with the responses to the reviewers below. The present paper is quite interesting and is suitable for the special issue "Liver Cancer: Molecular Mechanisms and Targeted Therapies".
The present paper is an original unpublished work, not submitted elsewhere, and together with the other authors, I declare that there are no other potential competing interests.
RESPONSES TO REVIEWERS
Reviewer 1
Comments and Suggestions for Authors
In this review, the authors investigated a variety of concomitant medications that could be counteracted because the efficacy of monotherapy with immune checkpoint inhibitors (ICIs) was not significant. This reviewer would like to raise some issues for the revision.
Paragraph 1: There is a very little description of introduction to ICIs. It would be better if the introduction part should include a comprehensive description of ICIs for chemo-therapeutics. Line 82: 7,6 months, 7.6 months. There are needing the same revisions throughout the article. line 91-92: The results of the PFS and OS on the untreated (or with another option for treatment) comparison should be presented. Paragraph 3: No information was given about the survival rate following CTLA-4. Paragraph 4: There is a lack of etiology information for the subjects in efficacy by combination with ICIs. It is advisable to state what pre-treatment the patient has undergone at a certain stage. Lines 145-152: No information on the clinical outcomes following FOLFOX4. It would be better if paragraph 4 will be divided into two groups. The first group is currently undergoing clinical progress and that can provide results for PFS, OS, and so on. The other group is under preclinical or early stage of clinical trials. Please discuss it in the conclusion or optimal part. Even if it's a palliative care targeting terminal cancer patients, it's always debated point whether it's judged that prolonging life about 2 months can have significant clinical significance.
The introduction has been improved with the inclusion of a description of ICIs for chemo-therapeutics.
Line 82 has been modified.
Line 91-92: the KEYNOTE-224 is a non-randomised, multicentre, open-label, phase 2 trial, no confront arm has been considered.
Paragraph 3: We have added the value of PFS in patients treated with the anti-CTLA-4 antibody namely Tremelimumab. Moreover, we added the recent results of another trial combining tremelimumab with the anti-PDL1 antibody Durvalumab.
Paragraph 4: We have added the clinical outcome concerning the use of FOLFOX4. Furthermore, we have added the information regarding the effectiveness mechanisms of the combinatorial approaches with ICIs. Regarding the possibility of a division of the paragraph in two group, this does not seem applicable because there are several types of combinatorial approaches.
Concerning the sentence” Even if it's a palliative care targeting terminal cancer patients, it's always debatable whether the principle that prolonging life about 2 months can have significant clinical significance.” we have added some considerations in the conclusions which consider the putative role of ICIs in the quality of life of HCC patients.
Thank you in advance for your consideration.
Antonella Argentiero, MD
Medical Oncology Unit
National Cancer Institute “Giovanni Paolo II”
Viale O. Flacco, 65 - 70124 Bari, ITALY
Phone/Fax: +39-0805555419
E-mail: argentieroantonella@gmail.com
Reviewer 2 Report
In this manuscript, the authors reviewed the role of immune checkpoint inhibitors in HCC.
Major remarks:
Nor cabozantinib, regorafenib, and ramucirumab are therapeutics against PD-L1/PD1. However, this is stated in the abstract. An overview figure is suggested of the receptors and therapeutic interactions. An overview table of finished/ongoing studies and outcome is strongly suggested. An overview table of the treatments and its mediators is strongly recommended. Extensive spelling and grammar revision is Please hypothesize more extensively on further improvements.
Minor remarks:
When abbreviations are mentioned for the first time they should be written to the full extent. Certainly in the abstract. Line72-73 requires proper citation. Atezolizumab and bevacizumab Is mentioned without a side note of its mechanism. Line 132 : ½ please adjust.
Author Response
October 07th, 2019
Prof. Dr. Edgaras Stankevičius
Editor-in-Chief
Dear Prof. Dr. Edgaras Stankevičius,
Please find attached our revised manuscript entitled: “Emerging role of Immune Checkpoint Inhibitors in Hepatocellular Carcinoma”.
To the best of our knowledge this is an interesting review reporting the role of immune checkpoint inhibitors in hepatocellular carcinoma.
We prepared the manuscript with the revision required and with the responses to the reviewers below. The present paper is quite interesting and is suitable for the special issue "Liver Cancer: Molecular Mechanisms and Targeted Therapies".
The present paper is an original unpublished work, not submitted elsewhere, and together with the other authors, I declare that there are no other potential competing interests.
RESPONSES TO REVIEWER
Reviewer 2
Comments and Suggestions for Authors
In this manuscript, the authors reviewed the role of immune checkpoint inhibitors in HCC.
Major remarks:
Nor cabozantinib, regorafenib, and ramucirumab are therapeutics against PD-L1/PD1. However, this is stated in the abstract.
We corrected the sentence in the abstract.
An overview figure is suggested of the receptors and therapeutic interactions. An overview table of finished/ongoing studies and outcome is strongly suggested. An overview table of the treatments and its mediators is strongly recommended.
We inserted a figure on the receptors and therapeutic interactions. In addition, we elaborated a single table including treatments and their mediators for each finished/ongoing studies and outcome.
Extensive spelling and grammar revision is
A certified English reviser corrected the text.
Please hypothesize more extensively on further improvements.
We expanded the discussion regarding the further improvements in conclusion as suggested.
Minor remarks:
When abbreviations are mentioned for the first time they should be written to the full extent. Certainly in the abstract.
We corrected the abbreviations in the abstract and in the text.
Line72-73 requires proper citation.
We provided the references as required
Atezolizumab and bevacizumab Is mentioned without a side note of its mechanism.
We inserted their mechanisms in the text.
Line 132 : ½ please adjust.
We corrected the mistyping.
Thank you in advance for your consideration.
Antonella Argentiero, MD
Medical Oncology Unit
National Cancer Institute “Giovanni Paolo II”
Viale O. Flacco, 65 - 70124 Bari, ITALY
Phone/Fax: +39-0805555419
E-mail: argentieroantonella@gmail.com

Round 2
Reviewer 1 Report
All concerns clearly addressed. This reviewer has a no additional issue to raise.